# Evaluation method on seismic risk of substation in strong earthquake area

**Jiawei Cui[1¤a], Ailan Che[1¤a]\*, Sheng Li[2¤b], Yongfeng Cheng[2¤b]**

1 Department of Civil Engineering, Shanghai Jiao Tong University, Shanghai, China, 2 Department of Electric Engineering, China Electric Power Research Institute, Xicheng District, Beijing, China

¤a Current address: Ocean and Civil Engineering, Shanghai Jiao Tong University, Shanghai, China
¤b Current address: China Electric Power Research Institute, Xicheng District, Beijing, China
\* alche@sjtu.edu.cn

**Data Availability Statement:** Our data set has been committed to a public repository and the relevant DOIs is https://datadryad.org/stash/dataset/doi:10.5061/dryad.8gtht76q6.

## Abstract

Frequent earthquakes in strong earthquake areas pose a great threat to the safety operation of electric power facilities. There exists a pressing research need to develop an assessment method for the seismic risk of substations, i.e., the hubs of power system networks. In this study, based on Incremental Dynamic Analysis (IDA), Probabilistic Seismic Demand Model (PSDM) and reliability theory, a vulnerability model for a substation is obtained, based on considering the relationships between Peak Ground Acceleration (PGA) and four seismic damage states (complete, extensive, moderate, and slight.) via a probabilistic approach. After an earthquake, the scope of influence and PGA distribution are evaluated using information recorded by the seismic observation stations, based on using interpolation or an empirical formula for the PGA attenuation. Therefore, the seismic risk can be evaluated by combining ground motion evaluation and the pre-built vulnerability model. The Wuqia- Kashgar area of Xinjiang was selected as the study area; it is an Earthquake-prone area, and one of the starting points for new energy transmission projects in China. Under a hypothetical earthquake ($M_S$ 7.9), the seismic risk of the substations was evaluated. The results show that: this method is able to give the probabilities of the four damage states of the substations, four substations close to the epicenter only have a probability of slight damage (45%-88%) and other substations are safer.

## 1. Introduction

A power system, is one of the main lifeline projects for a nation, society, and economy, and damage to a power system can cause unpredictable losses [1]. The safety of a substation is particularly important, as it acts as a hub of the power system [2,3]. The most destructive factor threatening a substation is undoubtedly an earthquake. For example, the 1976 Tangshan earthquake caused severe damage to the corresponding power system, making the city's lifeline projects unable to function normally, including those for the water supply, communications, combustion gas, medical aid, heat supply, firefighting, and material supply [4]. The 1989 Loma earthquake caused damage to three major substations at Moss Landing, Metcalf, and San Mateo in the San Francisco Bay area, respectively, resulting in power outages for 1.4 million

**Funding:** This work was part of the Program on 'Visualization of post-disaster information based on big data mining technology' and 'Study on on-line seismic monitoring and rapid damage evaluation method for substation (converter station).' funded under the Ministry of Science and Technology of the People's Republic of China, (grant number 2018YFC1504504 and 2018YFC0809404) awarded to AC. The project provides funding for research into how to quickly and effectively assess damage and visualize disaster information after an earthquake.

**Competing interests:** The authors have declared that no competing interests exist.

customers [5]. In the 2008 Wenchuan earthquake in China, there were more than 40 cases of leakage, seven cases of displacement, and 58 cases of casing damage in the transformers of 110 kV (and above) substations in the Sichuan Power Grid (Fig 1) [6]. Many studies have shown that major earthquakes may cause severe damage to substations; this will not only lead to the interruption of the main services of the power system, but will also affect earthquake relief work. Recently, ultra-high voltage transmission and transformation systems with bulk power grids, large generation units, high voltages and high automation have been gradually applied in power systems [7]. However, as the voltage level of a substation continues to increase, its vulnerability will also increase, and the consequences and losses caused by damage to it will become increasingly serious [8,9]. Therefore, it is very important to evaluate the seismic risk of substations after an earthquake; this can not only help reasonably allocate maintenance personnel and paths, but can also ensure the rapid restoration of power and smooth progress in disaster relief work.

In view of this, scholars have conducted many seismic studies on substations, mainly from two perspectives: seismic performance, and the vulnerability of the electrical equipment. Analyses of the seismic performances of substations have generally been performed based on numerical simulations [10–12] and physical modeling [13,14]. However, these studies have focused on the seismic design, isolation, and damping measures for power facilities before an earthquake. It is difficult to accurately and comprehensively assess the damage state of a substation after an earthquake. In contrast, the vulnerability analysis method uses probability to evaluate the seismic risk of a substation [15]. The vulnerability analysis method for power facilities is different from that for general civil construction and industrial equipment. Although there may be no major structural damage, the function may be seriously affected. A database addressing the seismic performances of substation equipment in California, jointly funded by Pacific Gas and Electric and Pacific Earthquake Engineering Center, contains relatively complete seismic damage data. The method employed constructs a relationship curve between the equipment damage rate and ground motion parameters through statistical analysis of historical earthquake damage data, i.e., a vulnerability model. Based on the damage to facilities of the Sichuan power grid in the Wenchuan earthquake, Hailei et al. and Mingpan used statistical regression methods to fit the vulnerability curves of various substation equipment, based on statistics from historical records of substation earthquake disasters [16,17]. Anagnos et al. drew a failure probability curve for substation equipment based on detailed data from the California substation earthquake disaster [18]. Although a seismic damage statistical analysis method is generally more accurate, it is difficult to obtain sufficient statistical data, and historical data are time-effective.

Another method of building the vulnerability model uses shaking-table testing and numerical simulations. The former is used to evaluate the dynamic characteristics of a structure, and the latter is used to obtain large amounts of data samples. Paolacci et al. conducted a vulnerability analysis for a high-voltage isolating switch [19]. Zareei et al. built a vulnerability model for a 420 kV circuit breaker with multivariate vulnerability [20]. Bender J et al. studied the vulnerability of a high-voltage bushing of a main transformer [21]. A vulnerability model can construct a relationship between seismic intensity and the seismic damage states of power facilities; moreover, the research on single power facilities is relatively mature. Therefore, it is possible to quickly evaluate the damage states of power facilities after an earthquake by combining the ground motion evaluations and vulnerability model. However, the design of a substation is redundant, and thus the destruction of a single device does not necessarily lead to the failure of the substation. The relative importance of each power equipment in a substation system is also different. Nevertheless, it is very important to study how to evaluate the seismic vulnerability of substations from an overall perspective.

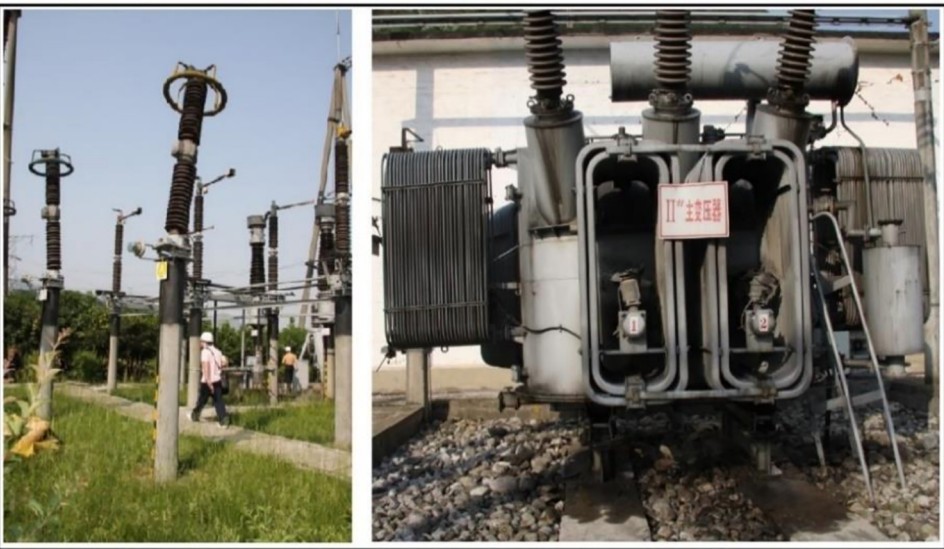

**Fig 1. Destruction of power facilities in the Wenchuan earthquake.**

This study presented herein is aimed at proposing a method for evaluating the seismic risk of a substation after an earthquake. The power facilities in the substation are divided into nine categories, and are assigned importance weights. A vulnerability model for substations with different voltage levels is constructed based on using an IDA, PSDM, and reliability theory in advance of an earthquake. When an earthquake occurs, the ground motion information, such as the epicenter, intensity, PGA, and distance to the epicenter can be quickly obtained through widely distributed seismic observation stations and a PGA attenuation law. Within the scope of seismic influence, the ground motion information of the substations is combined with the pre-built vulnerability model to complete seismic risk evaluations for substations with different voltage levels. A flowchart of this process is shown in Fig 2.

## 2. Evaluation of ground motion based on seismic observation stations

After an earthquake occurs, it is necessary to quickly obtain the PGA at the locations of the substations for a seismic risk assessment. However, not every substation has complete ground motion recording equipment, and even if so, the equipment are likely to be damaged during an earthquake. Widely distributed seismic observation stations can record ground motion information in real time. At present, most countries and regions worldwide have built complete networks of seismic observation stations. The United States Federal Emergency Management Agency has developed a disaster assessment system network including Shakemap, Hazards US, and 'Prompt Assessment of Global Earthquakes for Respons' systems, which can provide intensity information within 3–5 min after an earthquake. Japan has the "K-NET" and "KIKI-NET" seismic observation networks for quickly providing disaster investigations, quick intensity map reports, and other information. In Europe, multi-source observation methods such as those based on optics, light detection and ranging, and specific absorption rates are used to monitor earthquake disasters and analyze dynamic disaster conditions. By 2021, China will have 15,391 seismic observation stations, thereby ensuring that the spacing between stations in key surveillance areas is within 25 km. Therefore, the information recorded by seismic

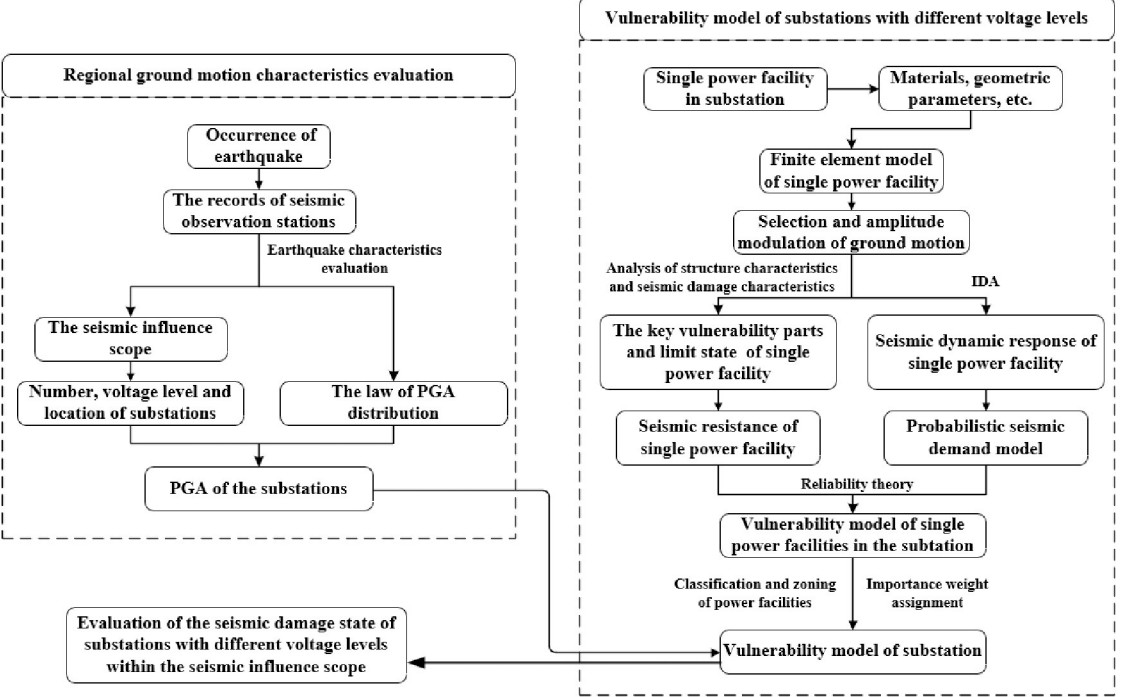

**Fig 2. Flowchart of seismic damage assessment of substation.**

observation stations can effectively provide ground motion inputs for seismic risk assessments of substations over a wide area.

However, there are many substations with different voltage levels over a large area and these substations do not necessarily coincide with seismic observation stations network. In order to obtain the PGA of the substation location after the earthquake, the distribution law of PGA can only be obtained by fitting the PGA data of the seismic network. In general, the attenuation model of PGA is difficult to consider the effect of site conditions. It is a complex process to clarify the influence of site conditions of hundreds of substations in a region on PGA in detail. In general, as with seismic observation stations, the foundation of the substation is reinforced and can be thought of as free ground shaking. The effect of ignoring the amplifying effects of soft soils can be considered small. Therefore, ignoring site conditions can ensure the efficiency of seismic risk assessment of substations, and will not affect the accuracy of probability-based assessment.

In view of the attenuation law of the PGA in the process of ground motion propagation, 'Evaluation of Earthquake Safety of Engineering Site' (GB17741-2005) provides recommended attenuation models, along with a unified expression as follows:

$$\lg(PGA) = C_1 + C_2M + C_3M^2 + (C_4 + C_5M)\lg[R + C_6\exp(C_7M)] \tag{1}$$

In the above, $M$ is the earthquake magnitude, and can be statistically regressed through the surface wave magnitude ($M_S$). $R$ is a distance parameter. $C_i(i = 1,2,\ldots,7)$ are the regression parameters. $C_6\exp(C_7M)$ is the near-field saturation factor, and is usually determined in advance according to the focal depth, and $C_3M^2$ is a large earthquake saturation factor. $C_2M$ is used to explain how high-frequency components often increase the acceleration in strong earthquake recording waves. In this study, the magnitude correlation factor and distance parameter correlation factor in the model were first decoupled, and then a distributed statistical regression was conducted to obtain the other parameters.

Eq (1) contains seven regression parameters. The results from a direct statistical regression still cannot meet the accuracy requirements for the free surface motion evaluation. Thus, it is necessary to propose further hypotheses and simplify the model. A commonly used method is to decouple the magnitude correlation factor and distance parameter correlation factor in the model, and then to perform statistical regression step-by-step to obtain other parameters. For a specific earthquake, the magnitude-related factors have little influence on the attenuation model. Therefore, the correlation factor for the distance parameter can be first calculated based on statistical regression, and the attenuation formula can be simplified as follows:

$$\lg(PGA) = C_8 + C_9\lg(R + C_{10}) \tag{2}$$

Here, $C_{10} = C_6\exp(C_7M)$. After fitting $C_8$, $C_9$, and $C_{10}$ based on historical earthquake information, other parameters can be obtained by further linear regression.

## 3. Vulnerability model for substations with different voltage levels

Earthquake disasters have evident regional characteristics, and there are often many different voltage substations within the scope of seismic influence. To evaluate the seismic risk, it is necessary to construct a vulnerability model. According to this paper [22], the vulnerability of a substation is determined based on the vulnerability of its power facilities. Therefore, assuming that each power facility is independent of the others, a vulnerability model for a substation can be obtained by combining these vulnerabilities. This section takes a 750 kV main transformer as an example to illustrate the construction process of the vulnerability model. The building process was as follows.

The power facilities in the substation were divided into nine categories, and corresponding importance weights were determined.

① A finite element model was built for each single power facility, and the natural frequency, natural period, key vulnerable parts, and limit state of each single power facility were analyzed.

② Ground motion records were selected for the numerical calculations, and an amplitude modulation was performed on them.

③ The vulnerability model for a single power facility was built based on IDA, the PSDM, and reliability theory.

④ The vulnerability model of the substation was built by assigning importance values to the power facilities of various categories.

### 3.1 Vulnerability model of single power facility: Case on 750 kV main transformer

**3.1.1 Structural properties and seismic damage characteristics of 750 kV main-transformer.**

(1) Nature characteristic:

The geometric dimensions and material parameters of a 750 kV main transformer can be obtained based on the "General design of power transmission and transformation project of State Grid." [23]. Then, the basic form of the finite element model is determined by extracting its geometric and material properties (Fig 3). The transformer comprises a main body, hoist seat, flange, low-voltage bushing, medium-voltage bushing, high-voltage bushing, and counterweight on busing top. The material and geometric parameters are listed in Table 1. Through

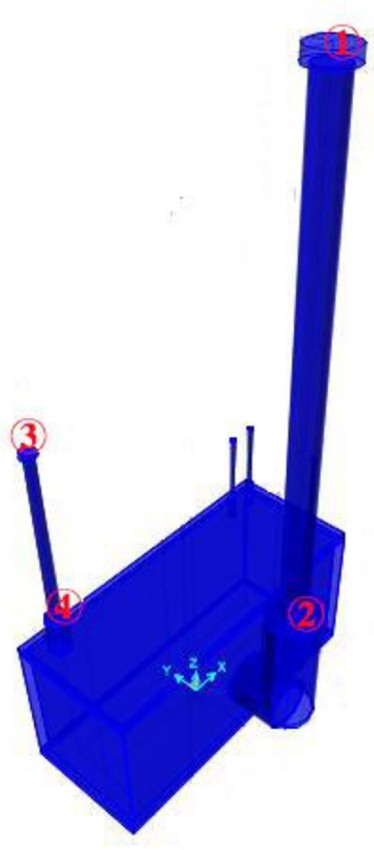

**Fig 3.** Finite element model of 750 kV main transformer and key vulnerability parameters (① the top displacement of the high-voltage bushing $U_g$; ② the maximum bending stress at the root of the high-voltage bushing $S_g$; ③ the top displacement of the medium-voltage bushing $U_z$; ④ the maximum bending stress at the root of the medium-voltage bushing $S_z$).

**Table 1. Parameters of 750 kV main transformer.**

| Number | Parameter | Value |
|--------|-----------|-------|
| 1 | The length of main body (m) | 6 |
| 2 | The width of main body (m) | 2.5 |
| 3 | The height of main body (m) | 3 |
| 4 | The weight of main body (kg) | 300000 |
| 5 | The elastic modulus of high-voltage bushing (GPa) | 100 |
| 6 | Poisson ratio of high-voltage bushing | 0.38 |
| 7 | The length of high-voltage bushing (m) | 8 |
| 8 | The outside diameter of high-voltage bushing (m) | 0.5 |
| 9 | The wall thickness of high-voltage bushing (m) | 0.05 |
| 10 | The weight of high-voltage bushing (kg) | 5000 |
| 11 | Equivalent stiffness coefficient of flange section | 0.5 |
| 12 | The ratio of medium-voltage bushing (for high-voltage) | 0.4 |
| 13 | The ratio of low-voltage bushing (for medium-voltage) | 0.4 |

**Table 2. Four limit states of 750 kV main transformer.**

| Limit state | Define |
|---|---|
| Complete damage | Impaired major functions. It may take a long time (e.g., 60 days) to repair |
| Extensive damage | Partial function interruption. It can be repaired over a period of time (e.g., 15 days) |
| Moderate damage | Some functions can be restored after a short interruption |
| Slight damage | Function play without interruption |

numerical simulation, the characteristic frequency is determined as 1.42, and the characteristic period is 0.7091 s.

(2) Key vulnerable parameters

Because the main body is heavy and has a fixed device at the bottom, a 750 kV main transformer rarely overturns and slips. However, a 750 kV main transformer contains composite insulator bushing. Owing to its mass concentration, large slenderness ratio, and brittle material, it can easily crack or break in the event of an earthquake [19]. Therefore, the bending stress at the root of the bushing and displacement of the top of the bushing are the limiting parameters. Four key vulnerable parameters are identified (Fig 3).

(3) Limit states:

In the vulnerability model for the power facilities, the limit states or seismic capacities of the key parameters refer to the quantitative description of four damage states: complete, extensive, moderate, and slight [24]. Table 2 defines the four seismic damage states corresponding to the 750 kV main transformer [25]. Based on the structural properties and material properties of the composite insulator bushing, it is considered that the bushing will undergo complete damage if the displacement of the casing exceeds 1/12 of the structural height and the maximum bending stress exceeds 40 MPa approximately. According to the literature [26], multiples of 0.7, 0.4, and 0.2 of the complete damage parameter values are selected for the extensive, moderate, and slight damage limit states, respectively (Table 3).

**3.1.2 Selection and treatment of ground motion.** To perform the IDA, a large number of ground motion time history records need to be selected. The selected ground motion records should have various applicability, and should not be restricted by the site. In addition, the discreteness of the ground motion records within a wide period range should also be considered, i.e., to ensure occasional uncertainty. Baker recommended a Pacific Earthquake Engineering Research (PEER) strong earthquake database ground motion record series [27]. Based on 'Code for seismic design of power facilities (GB 50260)', and 'Code for seismic design of buildings (GB50011)', the selection principle for the seismic waves for the IDA of 750 kV main transformer is shown in Table 4.

Then, 20 pieces of ground motion time history are identified from the PEER strong motion database according to the above conditions (Table 5).

**3.1.3 Incremental dynamic analysis (IDA) of 750 kV main transformer.**

(1) IDA method:

**Table 3. Parameter values of limit states of 750 kV main transformer.**

| Key vulnerability parameters | Parameter values of limit states | | | |
|---|---|---|---|---|
| | Complete | Extensive | Moderate | Slight |
| $U_g$ (mm) | 600 | 420 | 240 | 120 |
| $S_g$ (MPa) | 40 | 28 | 16 | 8 |
| $U_z$ (mm) | 200 | 140 | 80 | 40 |
| $S_z$ (MPa) | 40 | 28 | 16 | 8 |

**Table 4. Selection principle of ground motion records.**

| Parameter | Natural vibration period of 750 kV main transformer (s) | Venue category | Earthquake grouping | Earthquake influence coefficient |
|---|---|---|---|---|
| Attribute value | 0.7091 | 3 | 3 | 0.08 |

The IDA is a method for performing multiple dynamic time history analyses of a structural system. According to the calculation results, the dynamic response of the structural system can be obtained, and the results can be applied to a performance-based evaluation of the structure [28]. First, each selected ground motion record is multiplied by a series of scale factors for amplitude modulation, as shown in Eq (3). Subsequently, the multiple ground motion records after amplitude modulation are used for a dynamic analysis of the structure. Finally, the IDA curve is plotted to evaluate the performance of the structure.

$$a_\lambda = \lambda \cdot a \tag{3}$$

In the above, $a$ is the recorded acceleration value before amplitude modulation, and is shorthand for $a(t_i)$; $\lambda$ is the adjustment coefficient; and $a_\lambda$ is the recorded acceleration value after amplitude modulation.

According to the literature [29], when the earthquake intensity index is at the PGA, the amplitude levels shown in Table 6 can be used to perform the ground motion intensity amplitude modulation.

(2) Results of the IDA:

The characteristic period of 750 kV main transformer is 0.7019. Thus, the amplitude adjustment for the 20 ground motion data are 0.1 g, 0.2 g, 0.3 g, 0.4 g, 0.5 g, 0.6 g, 0.7 g, 0.8 g, 0.9 g, 1.0 g, 1.1 g, 1.2 g, 1.3 g, 1.4 g, and 1.5 g. In this way, a total of $20 \times 15 = 300$ ground motion records is obtained.

**Table 5. Pieces of information related to ground motion records.**

| Number | Time and Station Name | Joyner-Boore distance (km) | $M_S$ | PGA (g/10⁻²) |
|---|---|---|---|---|
| 1 | 1941 Ferndale City Hall | 91 | 6.6 | 4.53 |
| 2 | 1942 El Centro Array | 57 | 6.5 | 2.86 |
| 3 | 1951 El Centro Array | 25 | 5.6 | 5.90 |
| 4 | 1952 LA—Hollywood Stor FF | 115 | 7.36 | 5.33 |
| 5 | 1952 Pasadena—CIT Athenaeum | 123 | 7.36 | 4.48 |
| 6 | 1952 San Luis Obispo | 73 | 6 | 3.71 |
| 7 | 1953 El Centro Array | 15 | 5.5 | 4.39 |
| 8 | 1955 El Centro Array | 14 | 5.4 | 5.17 |
| 9 | 1956 El Centro Array | 121 | 6.8 | 5.14 |
| 10 | 1960 Ferndale City Hall | 57 | 5.7 | 5.97 |
| 11 | 1966 Cholame—Shandon Array | 18 | 6.19 | 4.72 |
| 12 | 1960 San Onofre—So Cal Edison | 129 | 6.63 | 4.15 |
| 13 | 1968 Via Tejon PV | 55 | 6.61 | 2.91 |
| 14 | 1971 LB—Terminal Island | 58 | 6.61 | 3.11 |
| 15 | 1971 Port Hueneme | 68 | 6.61 | 2.38 |
| 16 | 1971 Puddingstone Dam (Abutment) | 52 | 6.61 | 4.55 |
| 17 | 1971 San Juan Capistrano | 108 | 6.61 | 3.99 |
| 18 | 1971 UCSB—Fluid Mech Lab | 124 | 6.61 | 1.80 |
| 19 | 1971 Wheeler Ridge—Ground | 68 | 6.61 | 3.23 |
| 20 | 1971Wrightwood—6074 Park Dr | 61 | 6.61 | 4.55 |

**Table 6. Criterion for earthquake amplitude modulation.**

| Characteristic period of structure $T$(s) | Amplitude of PGA |
|---|---|
| $T \leq 0.5$ s | {0.1 g, 0.2 g, 0.3 g, . . ., 3.0 g} |
| $0.5$ s $< T \leq 2$ s | {0.1 g,0.2 g,0.3 g, . . ., 1.5 g} |
| $2$ s $< T$ | {0.1 g, 0.2 g, 0.3 g, 0.4 g, 0.5 g} |

Based on the dynamic time-history analysis method, a finite element analysis of 750 kV main transformer is conducted based on 300 ground motion records. Then, the dynamic response values of the four key vulnerable parameters of 750 kV main transformer can be plotted in Cartesian and logarithmic coordinate systems (Fig 4). These figures indicate that the seismic response values of four key vulnerability parameters ($U_g$, $S_g$, $U_z$, $S_z$.) under 20 sets of seismic waves, where the red line is the median of the seismic response values of 20 seismic waves under different PGA (0.1 g, 0.2 g, . . ., 1.5 g.). The bushing comprises a brittle material, and shows no evident deformation. Therefore, the shaded part in the figure indicates that the bushing has been completely destroyed.

**3.1.4 Vulnerability model of 750 kV main transformer based on probabilistic seismic demand model (PSDM).**

(1) Vulnerability model of each key vulnerable parameter based on PSDM:

The vulnerability model is built using the PSDM. The main purpose of the PSDM is to establish a probabilistic relationship between the intensity of ground motion and the seismic demand of a certain type of structure. Under the assumption that the seismic demand $D$ of the structure and peak ground motion $IM$ obey a lognormal distribution, the failure probability of the structure is expressed as follows [30]:

$$P[D \geq C/IM] = \Phi\left(\frac{\ln(IM) - \frac{\ln\bar{C} - \ln a}{b}}{\frac{\sqrt{\beta_{D/IM}^2 + \beta_C^2}}{b}}\right) \tag{4}$$

Here, $\bar{D}$ is the median value of the seismic response of structure; $\bar{C}$ is the median value of the structural seismic capacity, i.e., ultimate failure value of structure under earthquake action; $\beta_C$ is the standard deviation of the structural seismic capacity, in general, take 0.2; $\beta_{D/IM}$ is the log standard deviation of the structural seismic demand, and its median is $\ln\bar{D}$; its standard deviation is $\sigma$. $a$ and $b$ are unknown parameters that must be obtained through regression analysis. The value of $\beta_{D/IM}$, $a$, and $b$ can be estimated as follows:

$$\ln(\bar{D}) = b\ln(IM) + \ln(a) \tag{5}$$

$$\beta_{D/IM} \cong; \sqrt{\frac{\sum (\ln(d_i) - \ln(aIM_i^b))^2}{N-2}} \tag{6}$$

In the above, $d_i$ is the peak earthquake demand, and $IM_i$ is the peak ground motion. $N$ is the number of samples.

The seismic response results of each key vulnerability parameter from 3.1.3 was input into Eq (6) to calculate the $\beta_{D/IM}$ value, as shown in Table 7.

Then, according to the calculation results of the above sections and the limit states of each key vulnerable parameter, their vulnerability model can be obtained through Eq (4), as shown in Fig 5. These figures illustrate that the probability of four seismic states of key vulnerability parameters under different PGA. It can be seen that the seismic vulnerability of high-voltage

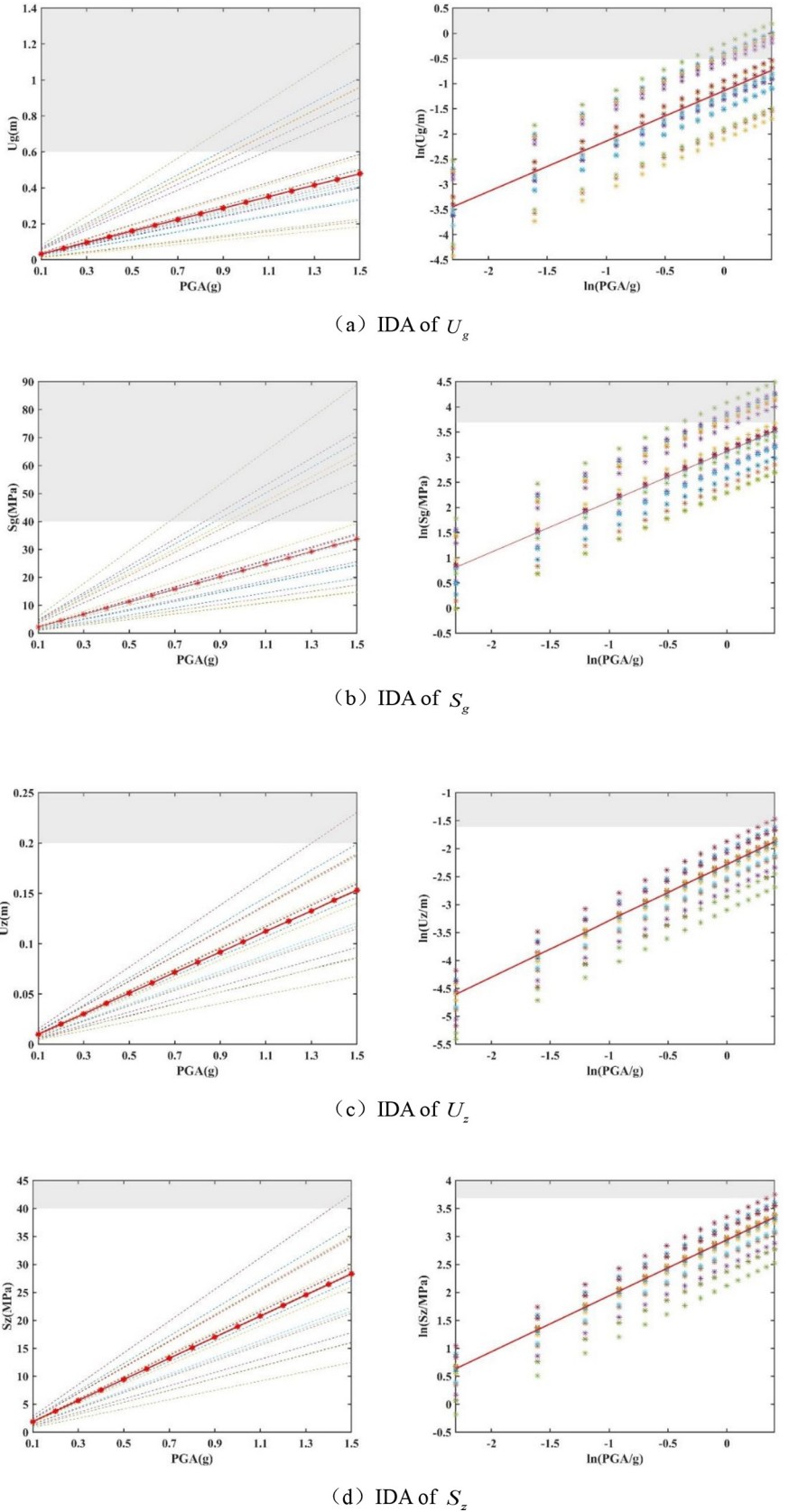

（a）IDA of $U_g$

（b）IDA of $S_g$

（c）IDA of $U_z$

（d）IDA of $S_z$

**Fig 4. Incremental dynamic analysis (IDA) of key vulnerable parameters.** (a) IDA of $U_g$. (b) IDA of $S_g$. (c) IDA of $U_z$. (d) IDA of $S_z$.

busing is higher than that of medium-voltage busing. High-voltage busing has a larger size than the medium, resulting in lower stiffness. As a result, high-voltage busing is more vulnerable to earthquakes. This shows the correctness of the analysis results from the side.

(2) Vulnerability model of 750 kV main transformer

After obtaining the vulnerability model for each key vulnerable parameter of the main transformer, it is necessary to determine the joint probability distribution of the different key vulnerable parameters reaching the limit state to construct the vulnerability model for the 750 kV main transformer. The seismic damage states between key vulnerable parameters are independent of each other, and their contributions to the overall functional loss of the system are the same. Therefore, the key vulnerable parameters are connected in series. Based on a simplified method of reliability theory, the reliability (probability of failure) of the series system is defined as follows [31]:

$$\max[P(E_i)] \leq P_{fs} \leq \left[\sum_{i=1}^{n} P(E_i), 1\right] \tag{7}$$

Here, $P(E_i)$ is the failure probability of a single critical factor, and $P_{fs}$ is the failure probability of the system. The lower boundary in Eq (7) represents the failure probability of the system when the failure events of the key vulnerable parameters are completely correlated, and the upper boundary represents the failure probability of the system when the failure events of each component are mutually exclusive. Furthermore, when the failure events of the key vulnerable parameters are statistically independent, the upper boundary can be rewritten as follows:

$$P_{fs} = 1 - \prod_{i=1}^{n} [1 - P(E_i)] \tag{8}$$

The probability curves of the three failure states for each key vulnerable parameter are substituted into Eqs (7) and (8). Then, the upper and lower boundaries of the four seismic damage state probabilities can be obtained for 750 kV main transformer; these comprise its vulnerability model (Fig 6). It can be seen that probabilistic seismic vulnerability analysis obtains the damage probability of the electrical equipment under different earthquake intensities, and quantitatively describes the seismic performance of the equipment. For example, when the intensity of the PGA reaches 1.0 g, the probabilities of the four damage states for the 750 kV main transformer are 100%, 97%–100%,15%–28%, and 0%, respectively.

## 3.2 Vulnerability model of substations with different voltage levels

**3.2.1 Classification of power facilities.** Because the relative importance and contributions of the different power facilities to the seismic damage of the substation are different and

**Table 7. Logarithmic standard deviation of each key vulnerable parameter.**

| Vulnerable parameters | Regression equation | Coefficient of determination $R^2$ | logarithmic standard deviation $\beta_{D/IM}$ | Correlation coefficient $\rho$ |
|---|---|---|---|---|
| $U_g$ | $\ln(U_g) = 1.0000\ln(PGA)+1.1434$ | 0.47 | 1.09 | 0.68 |
| $S_g$ | $\ln(S_g) = 1.0006\ln(PGA)+3.1125$ | 0.53 | 1.02 | 0.73 |
| $U_z$ | $\ln(U_z) = 1.0075\ln(PGA)+2.2827$ | 0.40 | 0.55 | 0.63 |
| $S_z$ | $\ln(S_z) = 1.0000\ln(PGA)+2.9382$ | 0.47 | 0.58 | 0.69 |

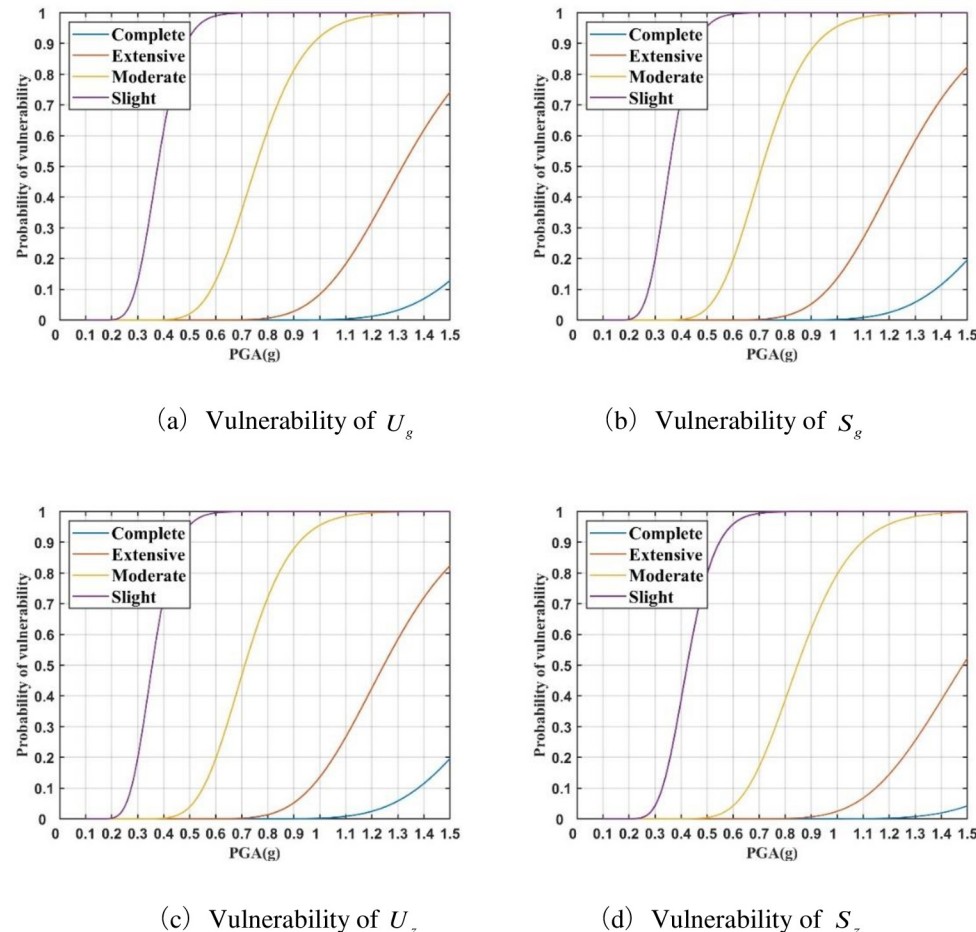

(a) Vulnerability of $U_g$  (b) Vulnerability of $S_g$

(c) Vulnerability of $U_z$  (d) Vulnerability of $S_z$

**Fig 5. Vulnerability models of four key vulnerable parameters.** (a) Vulnerability of $U_g$. (b) Vulnerability of $S_g$. (c) Vulnerability of $U_z$. (d) Vulnerability of $S_z$.

do not simply represent a series relationship, the vulnerability model for the entire substation cannot be calculated using Eqs (7) and (8). There are great differences in the layout of electrical equipment and structures in different substations, but they are relatively uniform in type.

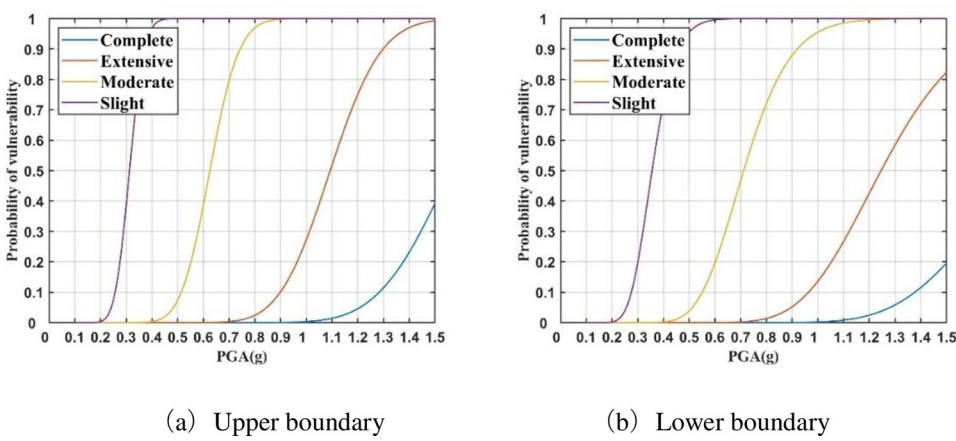

(a) Upper boundary  (b) Lower boundary

**Fig 6. Vulnerability model of 750 kV main transformer.** (a) Upper boundary. (b) Lower boundary.

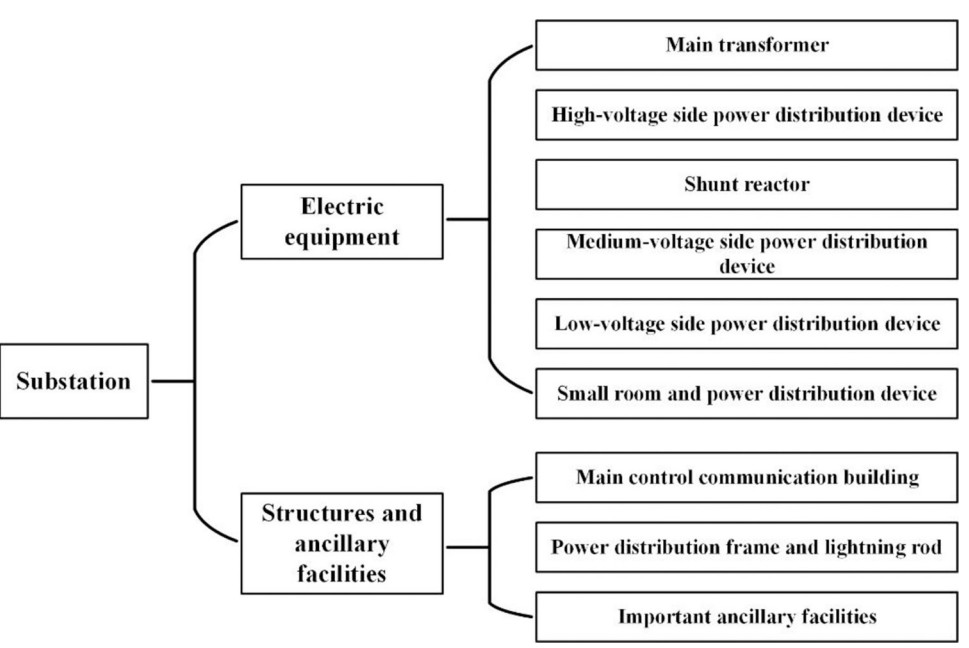

**Fig 7. Classification of power facilities in substation.**

According to this characteristic, in the study of the vulnerability of power facilities, typical types of power facilities can be selected to carry out example analysis, and on this basis, a universal applicability of substation seismic vulnerability analysis method can be formed.

This study adopts an importance weight assignment method to construct vulnerability models for substations with different voltage levels. The power facilities are classified based on their structural characteristics, seismic damage characteristics, and functions, as shown in Fig 7. Subsequently, the importance weight of each district power facility is obtained through referring to this criterion (Table 8) [25].

The vulnerability model for the substation can be obtained using Eq (9).

$$\sum_{i}^{n} \max(P_i(E_j)) \bullet F_i \leq P_{fw} \leq \sum_{i}^{n} \left[ 1 - \prod_{j}^{s} P_i(E_j) \right] \bullet F_i \tag{9}$$

In the above, $P_{fw}$ is the vulnerability of a substation, $P_i(E_j)$ is the vulnerability of a key part of a power facility, and $F_i$ is the importance weight of a zone-type power facility.

**Table 8. Earthquake safety risk assessment of substations-weights (F) of assessment items.**

| Number | Items | weight |
|:---:|:---:|:---:|
| 1 | Main control communication building | 0.18 |
| 2 | Main transformer | 0.19 |
| 3 | Shunt reactor | 0.15 |
| 4 | High-voltage side power distribution device (HVS) | 0.10 |
| 5 | Medium-voltage side power distribution device (MVS) | 0.09 |
| 6 | Low-voltage power distribution equipment and reactive power compensation (LVPD) | 0.08 |
| 7 | Small room and indoor power distribution device | 0.08 |
| 8 | Important ancillary facilities (such as fire protection) | 0.07 |
| 9 | Power distribution frame and lightning rod | 0.05 |

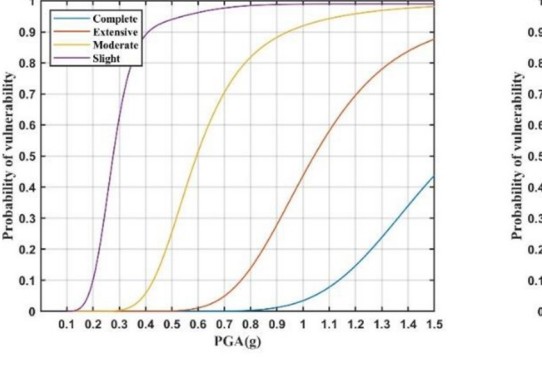
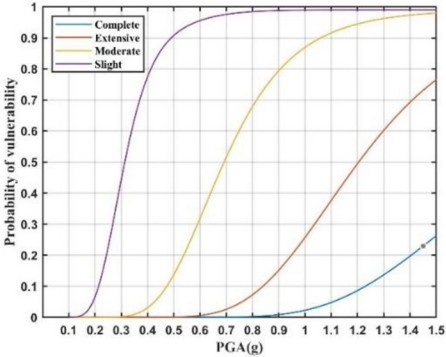

（a）Upper boundary （b）Lower boundary

**Fig 8. Vulnerability model of 750 kV substations.** (a) Upper boundary. (b) Lower boundary.

**3.2.2 Vulnerability model of substation with different voltage levels.** The vulnerability model for a substation can be constructed by assigning importance weights, as presented in Table 8, to the vulnerability curve of each district power facility. Based on the data provided by the 'General design of power transmission and transformation project of State Grid' [23], and Eq (9), vulnerability models are constructed for 100 kV, 220 kV, and 750 kV substations (Figs 8–10). It can be seen that these seismic vulnerability models can be used to construct the relationship between different ground motion intensities and seismic damage states of substations in a probabilistic way. Then, the seismic risk of substations within the affected area can be assessed after the earthquake.

# 4. Case study

## 4.1 Study area

The Wuqia-Kashgar area of Xinjiang was selected as the study area. It is located at 73.5˚ to 77.5˚ east longitude and 38.5˚ to 40.5˚ north latitude. This area is an earthquake-prone area, with complete historical seismic records. It has a widely distributed network of seismic observations and a complete record of historical earthquake information, facilitating the evaluation

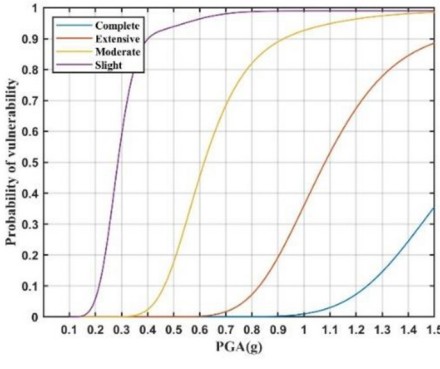
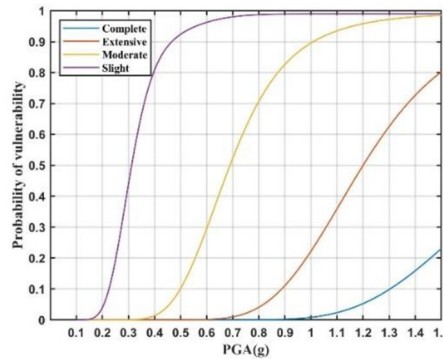

（a）Upper boundary （b）Lower boundary

**Fig 9. Vulnerability model of 220 kV substations.** (a) Upper boundary. (b) Lower boundary.

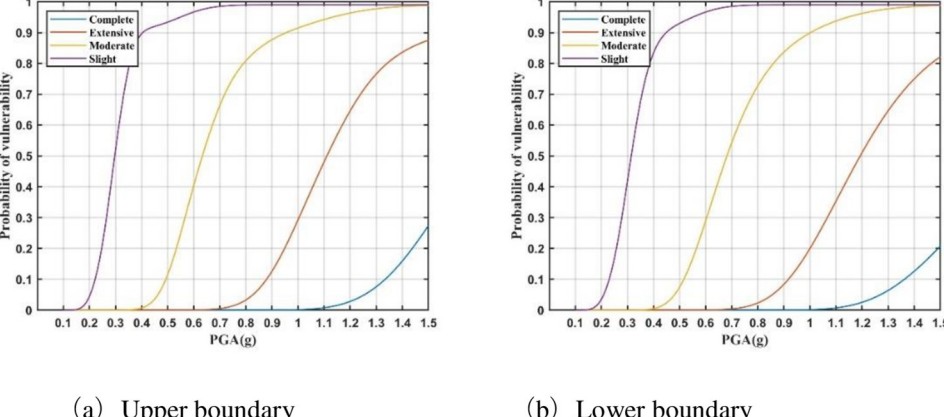

**Fig 10. Vulnerability model of 110 kV substations.** (a) Upper boundary. (b) Lower boundary.

of the ground motion characteristics. Fig 11(A) shows the earthquake catalog in the last 20 years, and Fig 11(B) shows the distribution of seismic observations in this area. Earthquakes of magnitude 2.9–6.8 have occurred 120 times.

Among them, the largest earthquake was the 2008 Wuqia $M_S$ 6.8 earthquake. The macro epicenter was located at 39°39' N, 73°52' E. Table 9 records the PGAs of each seismic observation station in Wuqia-Kashgar. Through Eqs (1) and (2), the PGA attenuation law of this earthquake was obtained as follows:

$$\lg(PGA) = 3.1376 - 0.6499 \times M + 0.1144 \times M^2 - 1.522\lg[R + 0.3736 \times \exp(0.5738 \times M)] \quad (10)$$

Subsequently, according to the actual situation after the earthquake, the results from Eq (10) were modified to obtain the intensity distribution diagram (Fig 12).

Notably, this area is one of the starting points for new energy (solar and wind power) transmission projects in China. It has a complete transmission system, and many substations that are threatened by earthquakes (Fig 13). Therefore, it was representative to choose this area for a research example.

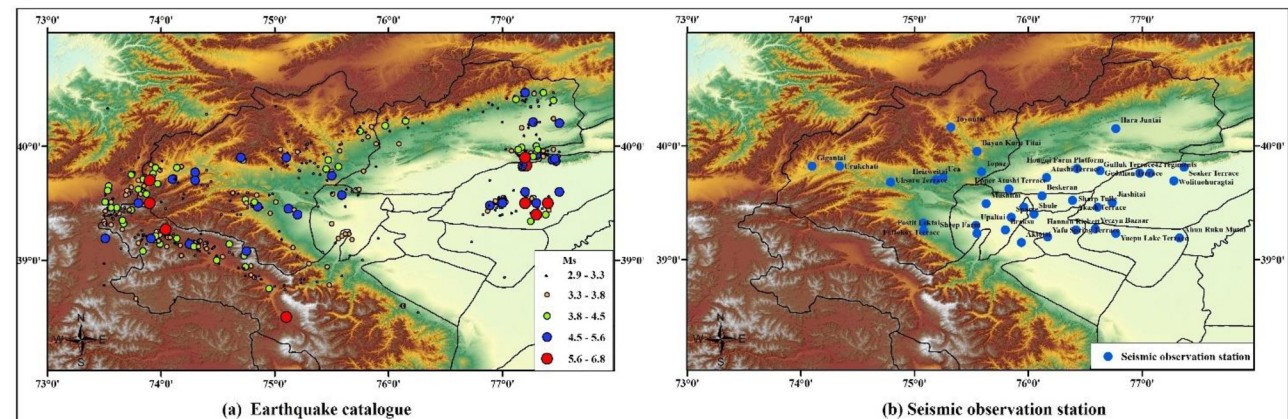

**Fig 11. Historical seismic information and seismic observation stations of the Wuqia-Kashgar area.** (a) Earthquake catalogue. (b) Seismic observation station.

**Table 9. Complete ground motion records of each station.**

| Number | Station name | Epicentral distance (km) | Horizontal PGA (g) | Vertical PGA (g) | The biggest PGA (g) |
|---|---|---|---|---|---|
| 1 | Akto | 201.2 | 0.0071 | 0.0027 | 0.0076 |
| 2 | Braksu | 186.5 | 0.0114 | 0.0035 | 0.0119 |
| 3 | Bayan kuruti | 165.4 | 0.0111 | 0.0050 | 0.0122 |
| 4 | Baishkran | 209.9 | 0.0078 | 0.0035 | 0.0085 |
| 5 | Gedalyan | 254.5 | 0.0109 | 0.0046 | 0.0118 |
| 6 | Gulluk | 284.3 | 0.0070 | 0.0029 | 0.0076 |
| 7 | Harajun | 272.4 | 0.0109 | 0.0039 | 0.0116 |
| 8 | Hongqi Farm | 237.3 | 0.0147 | 0.0053 | 0.0156 |
| 9 | Jiashi | 263.4 | 0.0074 | 0.0018 | 0.0076 |
| 10 | Jigan | 45.7 | 0.1273 | 0.0535 | 0.1381 |
| 11 | Finance | 197.7 | 0.0101 | 0.0036 | 0.0107 |
| 12 | Palltokoy | 124.4 | 0.0060 | 0.0061 | 0.0086 |
| 13 | 42 regiments | 295.8 | 0.0048 | 0.0015 | 0.0050 |
| 14 | Upper Artush | 184.8 | 0.0047 | 0.0034 | 0.0058 |
| 15 | Sparse | 197.7 | 0.0065 | 0.0039 | 0.0076 |
| 16 | Shule | 205.1 | 0.0115 | 0.0056 | 0.0128 |
| 17 | Toyun | 155.4 | 0.0115 | 0.0056 | 0.0128 |
| 18 | Topa | 165.5 | 0.0111 | 0.0046 | 0.0120 |
| 19 | Upal | 309.2 | 0.0053 | 0.0014 | 0.0055 |
| 20 | Uhsharu | 163.8 | 0.0087 | 0.0062 | 0.0107 |
| 21 | Uca | 96.3 | 0.0178 | 0.0111 | 0.0210 |
| 22 | Seaker | 136.1 | 0.0069 | 0.0047 | 0.0083 |
| 23 | Yuepu Lake | 317.5 | 0.0048 | 0.0020 | 0.0052 |
| 24 | Sheep farm | 269.4 | 0.0057 | 0.0019 | 0.0060 |

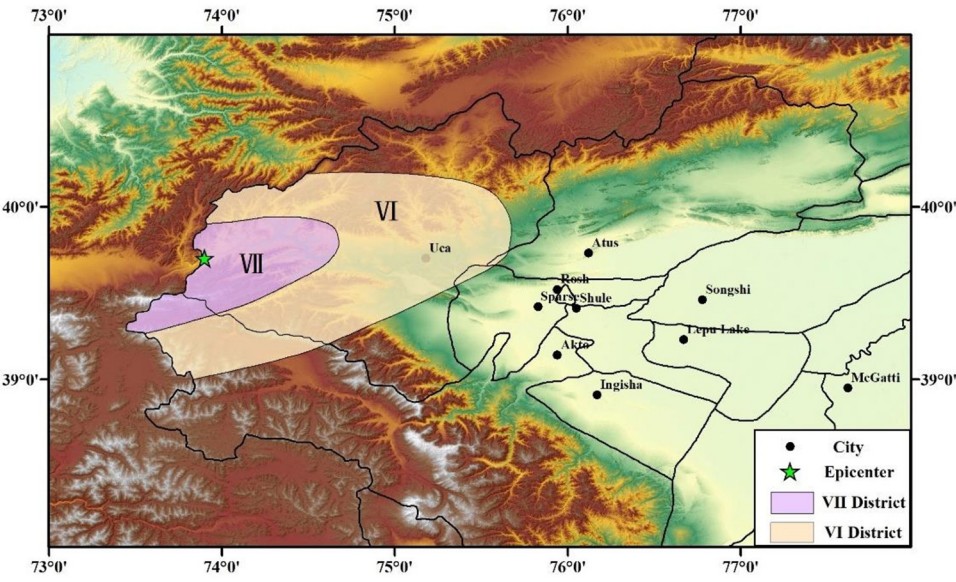

**Fig 12. Intensity distribution of the 2008 Wuqia M$_S$ 6.8 earthquake.**

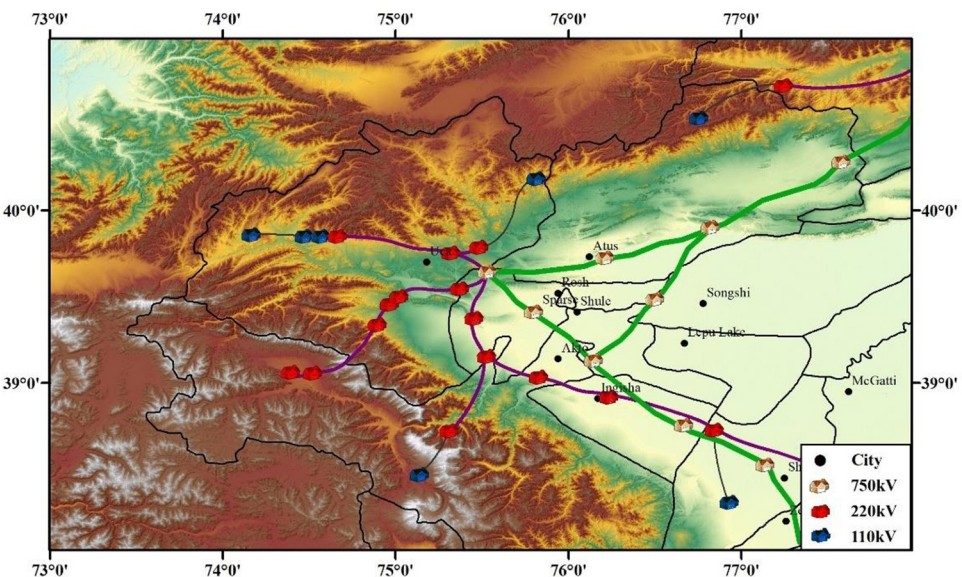

**Fig 13. Distribution of substations in Wuqia-Kashgar area.**

## 4.2 Evaluation of ground motion characteristics

Because there was no actual earthquake for supporting the study, this study assumed that when an earthquake at the upper magnitude limit of this area occurs, the seismic damage states of the substations are evaluated within the seismic influence scope. The evaluation process was as follows.

① The upper limit of the magnitude was calculated through the G-R model, assuming that its epicenter location, seismic influence scope, and PGA distribution law were the same as those of the 2008 Wuqia M 6.8 earthquake.

② The numbers, geographic locations, and voltage levels of substations within this area were determined, and their PGA values were calculated.

③ The seismic damage states of the substations were evaluated by inputting their PGAs into the pre-built vulnerability models.

**4.2.1 The upper limit of regional magnitude.**   The magnitudes and frequencies of earthquakes, whether global or regional earthquakes, generally satisfy Gutenberg-Rick's law, as shown in Eq (11) [32]. But the model does not take into account the upper limit of magnitude, as shown in Eq (12) [33]. In 1971, Utsu T proposed a modified G-R model that can take into account the upper limit of magnitude. Therefore, the modified G-R model was used to calculate the upper limit of the magnitude of the Wuqia-Kashgar earthquake.

$$\lg n(M) = a - bM \tag{11}$$

$$\lg n(M) = a - bM + \lg(M_c - M) \tag{12}$$

In the above, $n(M)$ is the number of earthquakes in the magnitude $M-\Delta M$ to $M+\Delta M$ range. The cumulative frequency is that $N(M) = \int_M^\infty n(M')dM'$. The value of $a$ describes the level of seismic activity in the selected area, and the value of $b$ is related to the regional medium

characteristics, stress state, and inhomogeneity, and can reflect the source $b$ characteristics of the earthquake. $M_c$ is the upper limit of regional magnitude.

For parameters $a$ and $b$, this study used the maximum likelihood method given by [33] to estimate their values. In the probability density function $f(X)$ of this model, $X = M - M_z$ ($M_z$ is the lower limit of the magnitude of the analysis data) is taken as the probability variable, and is given in Eq (13):

$$f(X) = B^2(C - B)\exp(-BX)/P \tag{13}$$

In the above, $\quad B = b\ln(10), \quad C = M_c - M_z, \quad P = \exp(-BC) + BC - 1.$ (14)

Then, the probability density function is expressed by two numbers, $B$ and $C$. If the number of earthquakes contained in a set of data is set as N, the logarithmic likelihood can be calculated by Eq (15).

$$\ln(L) = N\left(2\ln B - \ln P - B\sum_{i=1}^{N} X_i/N\right) + \sum_{i=1}^{N} \ln(C - X_i) \tag{15}$$

In Eq (14), the conditions for maximizing the logarithmic likelihood are that:

$$\frac{\partial}{\partial B}\ln L = 0, \quad \frac{\partial}{\partial C}\ln L = 0 \tag{16}$$

Thus, $B$ and $C$ can be obtained from Eqs (15), and (16). After that, $b$ and $M_c$ can be further solved by Eq (14).

The historical seismic information from the last 20 years, as shown in Fig 11(A), was taken as the data sample. The magnitude-cumulative frequency relationship was plotted as a curve, as shown in Fig 14. It can be seen that the cumulative frequency did not change within the magnitude range of 1.0–2.9. Therefore, it could not be included in the calculation range of the model. Between the magnitudes of 2.9, and 6.8, the linear relationship was good and the cumulative frequency distribution decreased with the increase in $M_S$, indicating that the data was complete and could be used for the model calculations.

According to the earthquake catalog data, the maximum likelihood method was used to fit Eqs (11)–(16) to obtain $M_c = 7.9$. So, the upper limit of regional magnitude ($M_S$) of the Wuqia-Kashgar earthquake is 7.9.

**4.2.2 The distribution of PGA.** It was assumed that the epicenter of the upper limit of the magnitude and the PGA distribution law were the same as those of the 2008 Wuqia $M_S$ 6.8 earthquake. Then, using Eq (10), the seismic influence scope and PGA distribution of this earthquake were obtained, as shown in Fig 15.

## 4.3 Evaluation of seismic damage state of substation

As shown in Fig 15, substations within the scope of the earthquake can be obtained through the State Grid Corporation of China, including the number, voltage level, distribution, and geographic location. There are 18 substations in the Wuqia-Kashgar area of Xinjiang, including three 750 kV substations, eleven 220 kV substations, and four 110 kV substations.

The PGA of each substation was substituted into the corresponding vulnerability model to obtain the seismic damage state, as shown in Fig 16. From the results, it can be seen that four substations close to the epicenter are threatened, and that the other substations are safer. Among them, owing to the distance from the epicenter, the 750 kV substations are barely affected, which is more important. For the 110 kV substation, which is close to the epicenter,

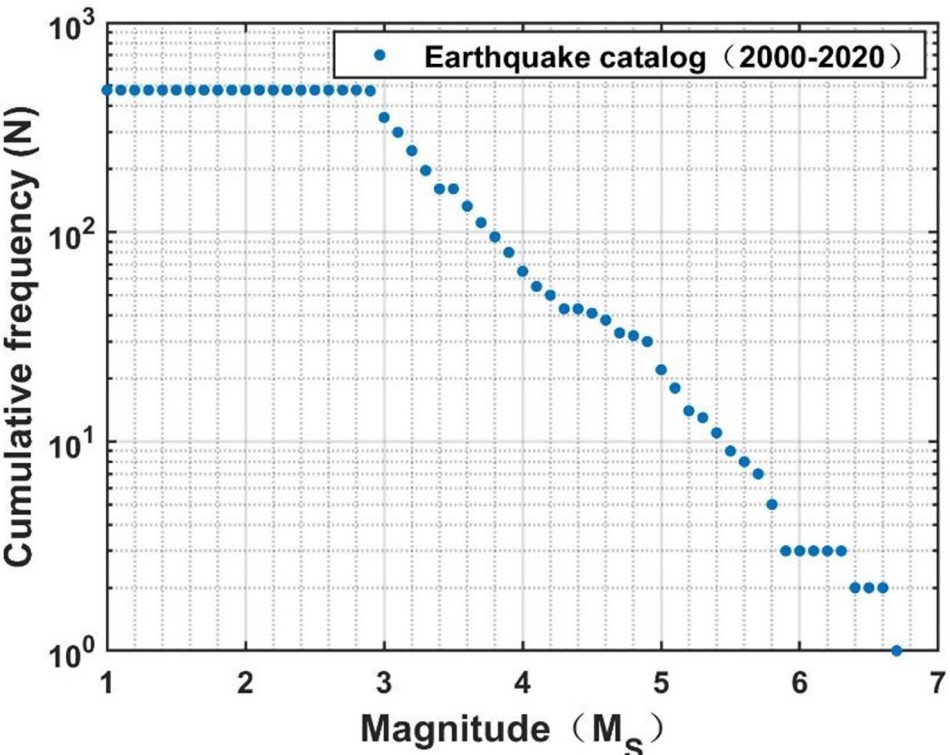

**Fig 14. Magnitude-cumulative frequency curve.**

the probabilities of the four damage states are 1%, 1%, 1%, and 82%-88%, respectively. The other three substations' (two 110 kV and one 220 kV) probabilities for the four damage states are 1%, 1%, 1%, and 45%-73%, respectively. The likelihood of major damage to these substations is unlikely, but slight damage is very likely, so repairs can be carried out subsequent to other priority repairs.

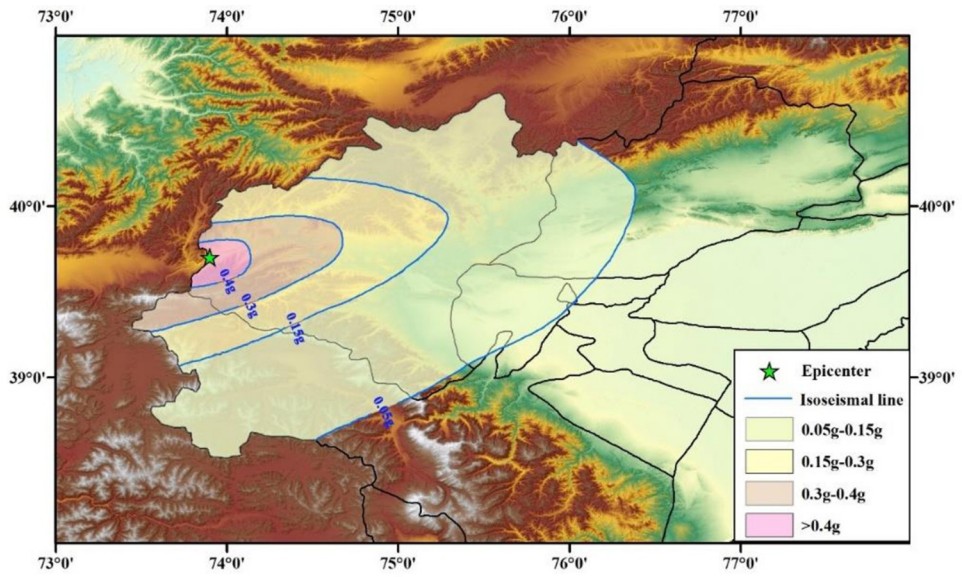

**Fig 15. Peak ground acceleration (PGA) distribution of the Wuqia-Kashgar area (assumed earthquake).**

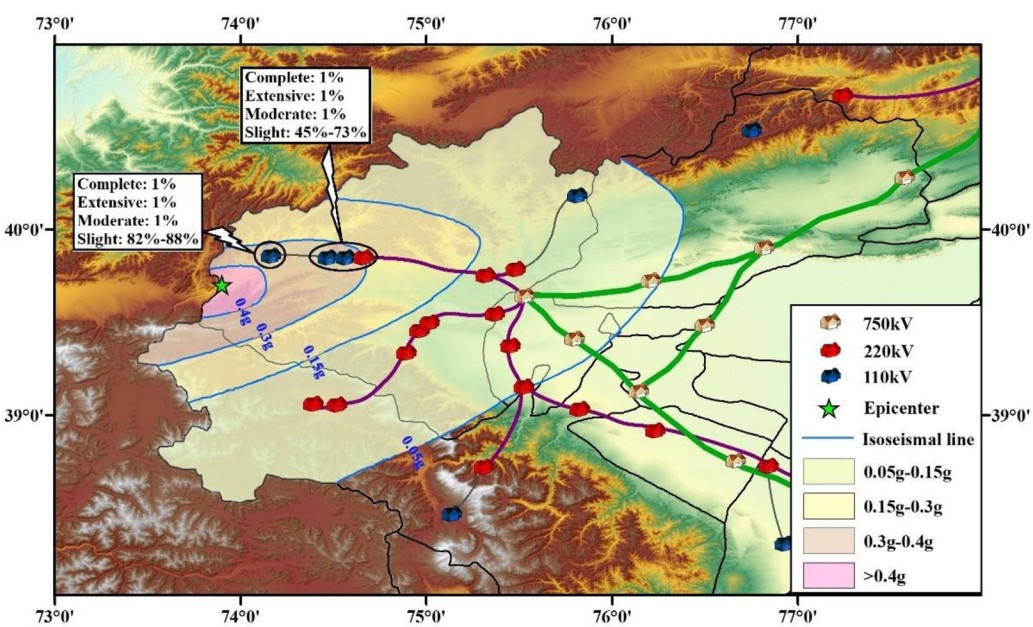

**Fig 16. Probability for four seismic damage states of substations.**

## 5. Conclusion

Considering the importance of substations after earthquakes, this study proposes a method for assessing the seismic risk of substations. This method combines widely distributed seismic observation stations and vulnerability models to rapidly evaluate the four damage states (complete, extensive, moderate, and slight) of substations in a probabilistic manner.

This method is applied to the Wuqia-Kashgar area of Xinjiang, which is an earthquake-prone area, and the starting point for new energy (solar and wind power) transmission projects in China. The conclusions are as follows.

① The vulnerability of a substation is determined by the vulnerability of the power facilities within the substation. Each substation is divided into nine categories of facilities, and the importance weights for each are given. Based on this, the relationship between the vulnerability of a single power facility and the overall vulnerability of the substation is constructed.

② The vulnerability models for the substations are able to give the probabilities of the four damage states after an earthquake.

③ Widely distributed seismic observation stations can provide sufficient data support for the rapid evaluation of the seismic damage states of substations.

④ The upper limit of the magnitude ($M_S$) for the Wuqia-Kashgar is 7.9. It is assumed that the epicenter location and PGA distribution of this earthquake are the same as that of the 2008 Wuqia $M_S$ 6.8 earthquake. Under this earthquake, the probabilities of occurrence of the four damage states for each substation within the seismic influence scope are determined.

Based on the above conclusions, it can be concluded that this method can effectively evaluate the seismic risk of substations, and provide a reference for managers to make decisions and provide services for earthquake relief work.

## Acknowledgments

The authors are grateful to all study participants.

## Author Contributions

**Data curation:** Sheng Li.

**Formal analysis:** Jiawei Cui.

**Investigation:** Jiawei Cui.

**Methodology:** Jiawei Cui, Ailan Che.

**Resources:** Yongfeng Cheng.

**Writing – original draft:** Jiawei Cui.

**Writing – review & editing:** Jiawei Cui.

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
