## [Decision Letter · Decision Letter 0]

26 Jul 2021

PONE-D-21-17828

Rapid evaluation method on seismic damage state of substation in strong earthquake area

PLOS ONE

Dear Dr. Che,

Thank you for submitting your manuscript to PLOS ONE. After careful consideration, we feel that it has merit but does not fully meet PLOS ONE’s publication criteria as it currently stands. Therefore, we invite you to submit a revised version of the manuscript that addresses the points raised during the review process.

The manuscript written English and fashion do not meet the standard of PLOS ONE. The manuscript needs substantial copyediting.Any abbreviation should be defined first, including those abbreviations used in the abstract i.e., incremental dynamic analysis (IDA), probabilistic seismic demand model (PSDM) and peak ground acceleration (PGA).Referencing style is inconsistent. In the text, the citation style is APA, however, in the reference section, they are sorted according to numbers. PLOS uses “Vancouver” style. Please follow Plos One referencing guidelines.Please correct and follow the Plos One author's guideline for submission of your revised manuscript.

We look forward to receiving your revised manuscript.

Kind regards,

Ahad Javanmardi, Ph.D

Academic Editor

PLOS ONE

Journal Requirements:

"This work is supported by the National Key R&D Program of China (2018YFC0809404)."

"This work is supported by the National Key R&D Program of China (2018YFC0809404)"

4. We note that Figure(s) 11, 12, 15, and 16  in your submission contain map images which may be copyrighted. All PLOS content is published under the Creative Commons Attribution License (CC BY 4.0), which means that the manuscript, images, and Supporting Information files will be freely available online, and any third party is permitted to access, download, copy, distribute, and use these materials in any way, even commercially, with proper attribution. For these reasons, we cannot publish previously copyrighted maps or satellite images created using proprietary data, such as Google software (Google Maps, Street View, and Earth). For more information, see our copyright guidelines: http://journals.plos.org/plosone/s/licenses-and-copyright.

1. You may seek permission from the original copyright holder of Figure(s) 11, 12, 15, and 16 to publish the content specifically under the CC BY 4.0 license.  

5. We note that Figure 1 in your submission contain copyrighted images. All PLOS content is published under the Creative Commons Attribution License (CC BY 4.0), which means that the manuscript, images, and Supporting Information files will be freely available online, and any third party is permitted to access, download, copy, distribute, and use these materials in any way, even commercially, with proper attribution. For more information, see our copyright guidelines: http://journals.plos.org/plosone/s/licenses-and-copyright.

2. If you are unable to obtain permission from the original copyright holder to publish these figures under the CC BY 4.0 license or if the copyright holder’s requirements are incompatible with the CC BY 4.0 license, please either i) remove the figure or ii) supply a replacement figure that complies with the CC BY 4.0 license. Please check copyright information on all replacement figures and update the figure caption with source information. If applicable, please specify in the figure caption text when a figure is similar but not identical to the original image and is therefore for illustrative purposes only

Reviewers' comments:

Reviewer's Responses to Questions

**Comments to the Author**

1. Is the manuscript technically sound, and do the data support the conclusions?

Reviewer #1: Yes

Reviewer #2: Yes

2. Has the statistical analysis been performed appropriately and rigorously? 

Reviewer #1: Yes

Reviewer #2: No

3. Have the authors made all data underlying the findings in their manuscript fully available?

Reviewer #1: No

Reviewer #2: No

4. Is the manuscript presented in an intelligible fashion and written in standard English?

Reviewer #1: Yes

Reviewer #2: No

5. Review Comments to the Author

Reviewer #1: Review Comments: This paper presents a methodology to assess the damage state of electrical substations after a seismic event provided that the vulnerability curve of the substation for different damage states is already developed. The approach seems good but the followings should be checked and modified:

The authors claim the method as rapid evaluation, but it's not like that as the vulnerability curve should be developed at first which includes incremental dynamic analysis (IDA), development of PSDM, application of reliability theory for the development of substation vulnerability curve from elemental vulnerability.

The manuscript should be checked thoroughly to modify several errors. These include-

a) Tables and Figures should be always addressed and described in text before they appear (e.g. Fig. 8, 9, 10 are not addressed or described in text).

b) Table No. and Figure No. described in text are not matched with actual Table No. and Figure No., respectively (e.g. it is described in text that Table 9 represents Records of PGA, but it is actually in Table 7; actually Figure 14 represents Magnitude-Frequency Curve while in text it is wrongly mentioned Figure 13.).

c) Reference list contains many styles that should be unified and some references should be checked, e.g. in line 174 though Baker is not included in the author list Lin et al. 2011, why author says "Baker recommended)?

Abstract:

As PGA depends not only on distance but also on soil significantly, interpolation for PGA attenuation will not give good or reliable results.

In the flowchart (Fig. 2), as the materials, geometric parameters, etc. are collected for the selected single facility; so, algorithm should be "single power facility > materials, geometric parameters > FEM....".

(3.1.1)

a) Is there any reference for the design manual mentioned in line 139?

b) In Table 1, Number 6- Poisson's ratio- It should not be in Pa rather unitless.

c) For finite element modeling seismic damage characteristics is not required, only geometric and material properties are needed (line 139-140).

d) It will be more suitable to define the four vulnerability parts as key vulnerability parameters (e.g. top displacement of high-voltage bushing, max principal stress at the root of medium-voltage bushing).

e) In Table 2 (which must be addressed in text), it seems that the damage limit states are defined as reverse. micro cracks are defined for complete damage while parts separated and bushing broke off completely are defined for slight damage.

f) As the height of medium-voltage bushing is significantly smaller than the high-voltage bushing, why same displacement limit (e.g. 600 mm for complete damage) is defined for two?

(3.1.2)

a. In Table 4, parameters are not defined clearly, e.g. attribute value 0.7019 refers to characteristic .....(what)?

[may be characteristic period, found later in line 197]

b. It should be clearly mentioned why earthuakes with very low PGA value (0.0180g - 0.0597g) are selected for IDA.

What is the type of magnitude of the earthquakes (Mw, Ms...)?

(3.1.3)

a) In line 197, it is mentioned that the amplitude adjustment ratios are 0.1g, 0.2g, 0.3g,....,1.5g. Is it amplitude adjustment ratios or amplitudes?

b) In Fig. 4, what is meant by [Ug/m], [Sg/MPa], [Uz/m], [Sz/m]?

c) In Fig. 4c, why >0.2m is marked while the limit is defined as 0.6m (in Fig. 4a >0.6m is marked for same limit state value)?

(3.1.4)

a) Parameters in equation 4 should be defined clearly

b) In line 229, though there is no section as 3.3.2 it mentioned as... simulation results from 3.3.2.

(3.2.1)

c) There is no description about the experts evaluation. The importance weights in Table 6 (addressed as Table 8 in text) should be checked.

(3.2.2)

d) What parameters are considered for nine different facilities and What are the limit state for those vulnerable parameters for different damage states?

(4.1)

e) The magnitude of 2008 Wuqia earthquake (Richter M 6.8) should be converted to Ms value as in the attenuation equation (eqn 1), it is denoted that magnitude should be in Ms.

(4.2.1)

a. In equation 10, N(>=M) denotes, rate of earthquake i.e. number of earthquakes per year with magnitude greater or equal M. This should be described correctly and also must be applied in the development of Magnitude-Frequency relationship.

b. The completeness of data can be checked by the plot of Cumulative No. of earthquakes Vs Time (Year) as per the simplified approach by Mulargia et al. (1987), or even by the method of Stepp (1972).

c. In the Magnitude-Frequency curve, vertical axis (earthquake frequency) should be plotted in log scale to represent the linear relationship.

d. It is not clear why the occurrence rate or frequency of maximum magnitude earthquake is 1, i.e. No. of earthquake per year is 1 [N(>=M)=1].

In conclusion it can be suggested that the article should be modified with application of recalculated values of various parameters as well as should be checked thoroughly to improve the write up.

Authors may consult with the following papers to improve the current state of the manuscript:

1. Hassanzadeh, R., Nedović-Budić, Z., Razavi, A. A., Norouzzadeh, M., & Hodhodkian, H. (2013). Interactive approach for GIS-based earthquake scenario development and resource estimation (Karmania hazard model). Computers & geosciences, 51, 324-338.

2. Okazaki, K., Villacis, C., Cardona, C., Kaneko, F., Shaw, R., Sun, J., ... & Tobin, L. T. (2000). RADIUS: Risk assessment tools for diagnosis of urban areas against seismic disasters. In RADIUS: Risk assessment tools for diagnosis of urban areas against seismic disasters (pp. 38-38).

3. Mazumder, R. K., & Salman, A. M. (2019). Seismic damage assessment using RADIUS and GIS: A case study of Sylhet City, Bangladesh. International journal of disaster risk reduction, 34, 243-254.

4. Applied Technology Council (ATC), ATC-13: Earthquake Damage Evaluation Data for California, Applied Technology Council, Washington, DC., 1985.

Reviewer #2: This paper study the seismic damage of the substation using rapid evaluation method. There is some question need to be clearly justified by the authors before it can be proceed for the next round.

Abstract need to rewrite as the current abstract too general. Please add the characteristic of substation, study area, number of ground motion etc. Most of the important info should be written in this section.

1. Why using PGA as the main IM in this study. As we know, Peak ground acceleration PGA, probably the most common IM today through which most of today's regulations define the design seismic forces (including EC8), was not indicated to be an efficient IM for both types of structural system and both types of soil. Most of the previous study found that, peak ground velocity PGV is nominated as a universal IM that could be used instead of the PGA. PGV obtained for all the analyzed cases, has almost twice smaller dispersion, which means four times the smaller number of earthquakes to achieve the same reliable estimation of seismic response. Also, the use of spectral response values Sa(T1), Sv(T1) and Sd(T1) has given very good results but we should bear in mind that their calculation is more complicated because they depend on the dynamic characteristics of the structure.

2. Need to add more recent references in this study.

3. In page 3, line 71, "..aimed at proposing a method for rapidly evaluating the damage state..". This is need further justify, either this study want to propose a method or to evaluate the damage state?.

4. In the introduction, author should highlight the research gap and new contribution in this field compare to other researcher.

5. Should have schematic diagram of the substation. Material used, strength etc.

6. Table 2 is suitable to use for substation?

7. Table 3 need to further justify about the limit state.

8. PEER is not an appropriate source of ground motion compare to COSMOS or other databases.

9. Justify the attribute value in page 8

10. What the main criteria in the selection of the ground motions as mentioned in Table 5.

11. Line 183 - 195 can be removed as this is well-known method in this field

12. Case study seem not reflect the main objective of this study. from my personal opinion, if we removed this section it wont give any affect to the findings and the main conclusion. Please justify.

13. In the title it mentioned about "rapid evaluation method" however the discussion about this method cannot be found in this study. Please justify.

13. Appropriate title should more related to "fragility assessment..."

6. PLOS authors have the option to publish the peer review history of their article (what does this mean?). If published, this will include your full peer review and any attached files.

Reviewer #1: No

Reviewer #2: No

---

## [Author Response · Author response to Decision Letter 0]

5 Sep 2021

List of Responses

Dear Editors and Reviewers:

Thank you for your letter and for the reviewers’ comments concerning our manuscript entitled “Rapid evaluation method on seismic damage state of substation in strong earthquake area” (which has been modified to ‘Evaluation method on seismic risk of substation in strong earthquake area’) (ID: PONE-D-21-17828). Those comments are all valuable and very helpful for revising and improving our paper, as well as the important guiding significance to our researches. We have studied comments carefully and have made correction which we hope meet with approval. Revised portion are marked in the paper. The main corrections in the paper and the responds to the reviewer’s comments are as following:

Responses to the editors:

Thank you for your letter and suggestions. According to the comments, we have made a comprehensive revision to the text and submitted the missing documents. It includes that: 1) The style and format of the text have been modified according to the template and requirements of the magazine; 2) Improved the sharpness of most figures; 3) Improved the English writing level of the text; 4) the text was revised according to the reviewers’ comments; 5) The owner of Fig 1 in the text is one of the authors (Yongfeng Cheng), we have submitted the document of the source of the picture; 6) A data availability statement, a funding statement, and the source and rightsholder information have been submitted. 7) Our data was submitted to a public database. DOI https://doi.org/10.5061/dryad.8gtht76q6. In addition, we would like to add two authors Yongfeng Cheng and Shen Li come from China Electric Power Research Institute They provided the following assistance in writing the text: 1) provide the data for the study; 2) provide the technical guidance; 3) the zoning and importance weights of power facilities are derived from the design manual and enterprise standards of China Electric Power Research Institute. Therefore, they are also one of the authors of the article. We hope we can get your approval.

Looking forward to hearing from you again!

Reviewer#1:

1: The authors claim the method as rapid evaluation, but it's not like that as the vulnerability curve should be developed at first which includes incremental dynamic analysis (IDA), development of PSDM, application of reliability theory for the development of substation vulnerability curve from elemental vulnerability 

Response: Thanks to you for your suggestions and opinions Considering the comments, we think that this method is more inclined to the seismic risk assessment of substations in the region, rather than the rapid assessment of earthquake damage state. The method proposed in this paper is to build the vulnerability model of each voltage level substation in advance and form a database. After the earthquake, the risk of the substation will be evaluated by matching the ground motion evaluation results with the vulnerability model. We have modified the title ‘Rapid evaluation method on seismic damage state of substation in strong earthquake area’ to ‘Evaluation method on seismic risk of substation in strong earthquake area’ and the presentation ‘rapid evaluation of seismic damage’ to ‘evaluation of seismic risk’ in the text to better match the proposed method.

2: Tables and Figures should be always addressed and described in text before they appear (e.g. Fig. 8, 9, 10 are not addressed or described in text).

Response: We are very sorry for our incorrect writing. We have adjusted the position of Tables and Figures (include Table 2) and added the description of Figures (include Fig 4, 5, 8, 9, 10). Ensures that all Tables and Figures are described in text before they appear. (Fig 4: line 215-2229) These figures indicate that the seismic response values of four key vulnerability parameters ( , , , ) under 20 sets of seismic waves, where the red line is the median of the seismic response values of 20 seismic waves under different PGA (0.1 g, 0.2 g, …, 1.5g). The bushing comprises a brittle material, and shows no evident deformation. Therefore, the shaded part in the figure indicates that the bushing has been completely destroyed. (Fig 5: line 252-261) These figures illustrate that the probability of four seismic states of key vulnerability parameters under different PGA. It can be seen that the seismic vulnerability of high-voltage busing is higher than that of medium-voltage busing. High-voltage busing has a larger size than the medium, resulting in lower stiffness. As a result, high-voltage busing is more vulnerable to earthquakes. This shows the correctness of the analysis results from the side. (Fig 8,9,10: line 309-320) It can be seen that these seismic vulnerability models can be used to construct the relationship between different ground motion intensities and seismic damage states of substations in a probabilistic way. Then, the seismic risk of substations within the affected area can be quickly assessed after the earthquake. 

3: Table No. and Figure No. described in text are not matched with actual Table No. and Figure No., respectively (e.g. it is described in text that Table 9 represents Records of PGA, but it is actually in Table 7; actually Figure 14 represents Magnitude-Frequency Curve while in text it is wrongly mentioned Figure 13.)

Response: We are very sorry for our carelessness in the writing The No. and descriptions of the Figures and Tables in text have been erratum. 

4: Reference list contains many styles that should be unified and some references should be checked, e.g. in line 174 though Baker is not included in the author list Lin et al. 2011, why author says "Baker recommended)

Response: We are sorry that an error occurred when writing the reference, which has been revised. In fact, the first author of this reference is Baker JW. In addition, the style of the references has been unified.？

(Line 502-503): [27] Baker, J W, Lin T, Shahi S K, Jayaram N. New ground motion selection procedures and selected motions for the PEER transportation research program. Pacific Earthquake Engineering Research Center, 2011.

Reviewer#2:

1: As PGA depends not only on distance but also on soil significantly, interpolation for PGA attenuation will not give good or reliable results.

Response: Thanks to you for your suggestions and opinions. We did not consider the effects of site conditions in our previous studies. Therefore, this effect has been reconsidered and discussed in the text. We added a discussion on why site conditions are not considered and explain the impact of that on the evaluation results. (Line 98-106): However, these substations do not necessarily coincide with seismic observation stations network. In order to obtain the PGA of the substation location after the earthquake, the distribution law of PGA can only be obtained by fitting the PGA data of the seismic network. In general, the attenuation model of PGA is difficult to consider the effect of site conditions. It is a complex process to clarify the influence of site conditions of hundreds of substations in a region on PGA in detail. In general, as with seismic observation stations, the foundation of the substation is reinforced and can be thought of as free ground shaking. The effect of ignoring the amplifying effects of soft soils can be considered small. Therefore, ignoring site conditions can ensure the efficiency of seismic risk assessment of substations, and will not affect the accuracy of probability-based assessment

2: In the flowchart (Fig. 2), as the materials, geometric parameters, etc. are collected for the selected single facility; so, algorithm should be "single power facility > materials, geometric parameters > FEM.....

Response: Thank you for your guidance. The flowchart (Fig 2) has been modified.

(3.1.1)

3: Is there any reference for the design manual mentioned in line 139

Response: This design manual was published by State Grid Corporation of China. Information about the book was added to the references.

(Line 494-495): [23] State Grid Corporation of China. General design of power transmission and transformation project of State Grid. Beijing, China: China Electric Power Press. 2017.

4: For finite element modeling seismic damage characteristics is not required, only geometric and material properties are needed (line 139-140).

Response: Considering the reviewer’s suggestion, the relevant expressions in text have been modified according to the advice of experts. (Line 151-152): Then, the basic form of the finite element model is determined by extracting its geometric and material properties (Fig 3).

5: It will be more suitable to define the four vulnerability parts as key vulnerability parameters (e.g. top displacement of high-voltage bushing, max principal stress at the root of medium-voltage bushing).

Response: Thanks for your suggestion. In the full text, 'key vulnerable Parts' has been changed to' key vulnerable parameter '.

6: In Table 2 (which must be addressed in text), it seems that the damage limit states are defined as reverse. micro cracks are defined for complete damage while parts separated and bushing broke off completely are defined for slight damage.

Response: We are very sorry for our incorrect writing. The seismic damage state of power facilities defined in Table 2 is based on the enterprise standard, which is ‘Guidelines for seismic safety risk assessment of electrical substations (converter stations)’, issued by the State Grid Corporation of China. Table 2 has been errata and added references.

(Line 498-499): [25] State Grid Corporation of China. Guidelines for seismic safety risk assessment of electrical substations (converter stations). Beijing, China: China Electric Power Press. 2021.

7: As the height of medium-voltage bushing is significantly smaller than the high-voltage bushing, why same displacement limit (e.g. 600 mm for complete damage) is defined for two?

Response: We are very sorry for our incorrect writing. The displacement of the top medium-voltage bushing should be 200mm for complete damage. 200mm was also used in subsequent analysis. So, errata have been performed on Table 3.

(3.1.2)

8: In Table 4, parameters are not defined clearly, e.g. attribute value 0.7019 refers to characteristic (what)? [may be characteristic period, found later in line 197]

Response: 0.70 is natural vibration period of 750 kV main transformer. More specific descriptions of these parameters in Table 4 have been added. At the same time, the criteria by which these parameters are used have been added. (Line 186-188): Based on ‘Code for seismic design of power facilities (GB 50260)’, and ‘Code for seismic design of buildings (GB50011)’, the selection principle for the seismic waves for the IDA of 750 kV main transformer is shown in Table 4.

9: It should be clearly mentioned why earthuakes with very low PGA value (0.0180g - 0.0597g) are selected for IDA. What is the type of magnitude of the earthquakes (Mw, Ms...)?

Response: Thanks for your suggestions. These seismic waves searched in the PEER strong earthquake database were recorded by seismic stations at some distance from the epicentre. Therefore, their PGA value is relatively small. After the corresponding amplitude modulation, it can be used for numerical calculation. The authors have revised the relevant statements in Table 5 and added Rjb (Joyner-Boore distance) for these seismic stations. The type of magnitude of the earthquakes is MS.

(3.1.3)

10: In line 197, it is mentioned that the amplitude adjustment ratios are 0.1g, 0.2g, 0.3g,....,1.5g. Is it amplitude adjustment ratios or amplitudes?

Response: We are very sorry for the writing is not clear enough. They are amplitudes not amplitude adjustment ratios. Each seismic wave is modulated into 15 seismic waves with PGA of 0.1-1.5g. The author revised the relevant statements in the article and corrected them.

11: In Fig. 4, what is meant by [Ug/m], [Sg/MPa], [Uz/m], [Sz/m]?

Response: We are very sorry for the lack of detail. We have added a description of these parameters in the text. (Line 215-219): These figures indicate that the seismic response values of four key vulnerability parameters ( , , , ) under 20 sets of seismic waves, where the red line is the median of the seismic response values of 20 seismic waves under different PGA (0.1 g, 0.2 g, …, 1.5g). The bushing comprises a brittle material, and shows no evident deformation. Therefore, the shaded part in the figure indicates that the bushing has been completely destroyed.

12: In Fig. 4c, why >0.2m is marked while the limit is defined as 0.6m (in Fig. 4a >0.6m is marked for same limit state value)?

Response: We are very sorry for our incorrect writing. Fig 4c is the seismic response calculation result of displacement at the top of medium-voltage busing (UZ). In the unmodified version, the limit value of UZ in Table 3 should be 0.2m, not 0.6m. The authors have made changes to Table 3.

(3.1.4)

13: Parameters in equation 4 should be defined clearly.

Response: We are very sorry for the lack of rigorous writing. We have added the description of each parameter in Eq. 4. 

14: In line 229, though there is no section as 3.3.2 it mentioned as... simulation results from 3.3.2.

Response: We are very sorry for our incorrect writing. It should be 3.1.3, not 3.3.2. We have added relevant content, explaining in detail how the results of IDA analysis in 3.1.3 applied to 3.1.4. (Line 247-249): The seismic response results of each key vulnerability parameter from 3.1.3 was input into Eq. (5) to calculate the value, as shown in Table 7.

(3.2.1)

15: There is no description about the experts evaluation. The importance weights in Table 6 (addressed as Table 8 in text) should be checked.

Response: We are very sorry for our unclear writing. Power facility classification and zoning and importance weights are cited from the enterprise standard, i.e. ‘Guidelines for seismic safety risk assessment of electrical substations (converter stations)’. This book was compiled by the two presenteers of this article, Shanghai Jiao Tong University and China Electric Power Research Institute. Three authors of the article participated in the compilation process. The author has marked references to the weight of importance in the text. (Line 294-297): This study adopts an importance weight assignment method to construct vulnerability models for substations with different voltage levels. The power facilities are classified based on their structural characteristics, seismic damage characteristics, and functions, as shown in Fig 7. Subsequently, the importance weight of each district power facility is obtained through referring to this criterion [25].

(Line 498-499): [25] State Grid Corporation of China. Guidelines for seismic safety risk assessment of electrical substations (converter stations). Beijing, China: China Electric Power Press. 2021.

(3.2.2)

16: What parameters are considered for nine different facilities and What are the limit state for those vulnerable parameters for different damage states?

Response 16: Due to the limitation of space, the analysis process and results of all power facilities cannot be included in the text. Other different facilities have been submitted to the data availability declaration.

(4.1)

17: The magnitude of 2008 Wuqia earthquake (Richter M 6.8) should be converted to Ms value as in the attenuation equation (eqn 1), it is denoted that magnitude should be in Ms.

Response: China officially uses the surface wave (MS) magnitude for earthquakes. All the earthquakes in the text are of surface wave magnitude. Due to an oversight in writing, the problem was not taken into account. The authors have unified the expression of the text and examined the data.

(4.2.1)

18: In equation 10, N(>=M) denotes, rate of earthquake i.e. number of earthquakes per year with magnitude greater or equal M. This should be described correctly and also must be applied in the development of Magnitude-Frequency relationship.

Response: We are very sorry for our unclear writing. Thanks to the expert who pointed out the mistakes in our paper. The previously used method could not get the upper limit of the magnitude of the region, so I used Mabuti Hiroshi’ (2004) method to get the solution again. 

19: The completeness of data can be checked by the plot of Cumulative No. of earthquakes Vs Time (Year) as per the simplified approach by Mulargia et al. (1987), or even by the method of Stepp (1972).

Response: Considering the reviewers’ suggestion, we have checked the completeness of data. These are all the magnitudes recorded in this region since observations began.

20: In the Magnitude-Frequency curve, vertical axis (earthquake frequency) should be plotted in log scale to represent the linear relationship.

Response: We are very sorry for our incorrect writing. We have modified the Fig 14.

21: It is not clear why the occurrence rate or frequency of maximum magnitude earthquake is 1, i.e. No. of earthquake per year is 1 [N(>=M)=1].

Response: Thanks to the expert for pointing out my mistake. It is wrong to think that the magnitude whose cumulative frequency is 1 is the upper limit for the region. There were errors in the previous method, and now we used a modified G-R model to analyze the upper limit of magnitude in the region. The upper limit of magnitude (MS) by using the modified G-R model.

---

## [Decision Letter · Decision Letter 1]

6 Oct 2021

Evaluation method on seismic risk of substation in strong earthquake area

PONE-D-21-17828R1

Dear Dr. Che,

We’re pleased to inform you that your manuscript has been judged scientifically suitable for publication and will be formally accepted for publication once it meets all outstanding technical requirements.

Kind regards,

Ahad Javanmardi, Ph.D

Academic Editor

PLOS ONE

Additional Editor Comments (optional):

Please correct the word "**Peek**" in the abstract. It should be **Peak** Ground Acceleration (PGA). 

Reviewers' comments:

Reviewer's Responses to Questions

**Comments to the Author**

1. If the authors have adequately addressed your comments raised in a previous round of review and you feel that this manuscript is now acceptable for publication, you may indicate that here to bypass the “Comments to the Author” section, enter your conflict of interest statement in the “Confidential to Editor” section, and submit your "Accept" recommendation.

Reviewer #1: All comments have been addressed

Reviewer #2: All comments have been addressed

2. Is the manuscript technically sound, and do the data support the conclusions?

Reviewer #1: Yes

Reviewer #2: Yes

3. Has the statistical analysis been performed appropriately and rigorously? 

Reviewer #1: Yes

Reviewer #2: Yes

4. Have the authors made all data underlying the findings in their manuscript fully available?

Reviewer #1: Yes

Reviewer #2: No

5. Is the manuscript presented in an intelligible fashion and written in standard English?

Reviewer #1: Yes

Reviewer #2: Yes

6. Review Comments to the Author

Reviewer #1: Thank you to the authors for considering my comments for improving their manuscript. The authors have addressed all of my comments satisfactorily. I have no further comments.

Reviewer #2: (No Response)

7. PLOS authors have the option to publish the peer review history of their article (what does this mean?). If published, this will include your full peer review and any attached files.

Reviewer #1: No

Reviewer #2: No

---

## [Editor Report · Acceptance letter]

10 Nov 2021

PONE-D-21-17828R1 

Evaluation method on seismic risk of substation in strong earthquake area 

Dear Dr. Che:

I'm pleased to inform you that your manuscript has been deemed suitable for publication in PLOS ONE. Congratulations! Your manuscript is now with our production department. 

Kind regards, 

on behalf of

Dr. Ahad Javanmardi 

Academic Editor

PLOS ONE